# Fatty liver index as a predictive marker for the development of diabetes: A retrospective cohort study using Japanese health check-up data

Atsushi Kitazawa[1,2]*, Shotaro Maeda[1], Yoshiharu Fukuda[1]

**1** Graduate School of Public Health, Teikyo University, Tokyo, Japan, **2** Department of Nephrology, Dokkyo Medical University Saitama Medical Center, Saitama, Japan

* sph-akitazawa@med.teikyo-u.ac.jp

**Data Availability Statement:** Data cannot be shared publicly because of agreement between data holder. Data are available from The Mutual Aid Association of Prefectural Government Personnel

## Abstract

### Background & aims

Fatty liver is associated with incident diabetes, and the fatty liver index (FLI) is a surrogate marker for non-alcoholic fatty liver disease (NAFLD). We aimed to determine whether or not FLI was associated with incident diabetes in relation to obesity and prediabetic levels in the general Japanese population.

### Methods

This was a retrospective study using the Japanese health check-up database of one health insurance from FY2015 to FY2018. This study included 28,991 individuals with prediabetes. First, we stratified all participants into two groups: "high-risk," comprising patients with HbA1c >6.0%, and "standard," comprising the rest. Subsequently, we divided them into four groups according to FLI (<30 or not) and obesity (BMI <25 kg/m$^2$ or not). Subsequently, the incidence rate of diabetes was compared among the groups after 3 years of follow-up using multiple logistic regression models after adjusting for potential confounders.

### Results

After 3 years of follow-up, 1,547 new cases of diabetes were found, and the cumulative incidence was 2.96% for the standard group and 26.1% for the high-risk group. In non-obese individuals, odds ratios (95% confidence interval) for FLI ≥30 versus FLI <30 were: 1.44 (1.09–1.92) for the standard group and 1.42 (0.99–2.03) for the high-risk group. In the high-risk group, obesity (BMI ≥25 kg/m$^2$) but FLI <30 was not a risk factor for developing diabetes.

### Conclusion

Although high FLI is generally considered to be a risk factor for developing diabetes, obesity might have been a confounding factor. However, the present study showed that high FLI is a risk factor for the development of diabetes, even in non-obese individuals. Our results include suggestion to develop a screening tool to effectively identify people at high risk of

(contact via https://www.chikyosai.or.jp/) for researchers who meet the criteria for access to confidential data.

**Funding:** The authors received no specific funding for this work.

**Competing interests:** The authors have declared that no competing interests exist.

developing diabetes from the population (especially non-obese prediabetes) who are apparently at low health risk and are unlikely to be targeted for health guidance.

## Introduction

Non-alcoholic fatty liver disease (NAFLD) refers to a fatty liver without a history of excessive alcohol consumption or liver disease. It is becoming a common chronic liver disease worldwide. Its prevalence in the general population is reported to be approximately 25% worldwide [1] and 18%–30% in Japan [2–4]. It is considered to be a hepatic phenotype of metabolic syndrome (MetS) and is closely related to obesity. Recent meta-analysis studies have shown that patients with NAFLD are approximately twice as likely to develop diabetes as those without NAFLD [5].

Liver biopsy is the gold standard for diagnosing NAFLD. Since it is an invasive procedure, abdominal ultrasonography is clinically used for the diagnosis. In 2006, Bedogni et al. introduced the fatty liver index (FLI) as a surrogate marker for NAFLD, which comprises the body mass index (BMI), waist circumference (WC), and gamma-glutamyl transferase (GGT) and triglyceride (TG) levels [6]. Movahedian et al. performed a meta-analysis of NAFLD defined by FLI (FLI-NAFLD) and the risk of developing diabetes and concluded that high FLI scores increased the risk of developing diabetes [7].

Cases of most previous studies on the association between FLI and the development of diabetes have been a mixture of normoglycemia and prediabetes, or "not diabetic." However, normoglycemia and prediabetes might differ in the risk of diabetes and related factors. Heianza et al. reported that patients with prediabetes diagnosed based on impaired fasting glucose (IFG; fasting plasma glucose (FPG) $\geq$100 mg/dL) and/or hemoglobin A1c (HbA1c) ($\geq$5.7%) according to ADA criteria [8] are six times more likely to develop diabetes than patients with normoglycemia [9]. In evaluating the risk of developing diabetes, subjects should be differentiated between normoglycemia and prediabetes.

In addition, only 20% of patients with prediabetes met both criteria (IFG and HbA1c) in the study by Heianza et al. [9]. Prediabetes should be extracted with criteria for both HbA1c and IFG because applying only one criterion often leads to a missed diagnosis. Among previous studies on prediabetes [10–13], only Nadal et al. extracted prediabetes using both HbA1c and FPG criteria [11].

Even in the prediabetic population, the risk of developing diabetes is strongly affected by an individual's glucose tolerance level. Therefore, the statistical analysis should be performed after adjusting for baseline glucose tolerance levels. Of these four studies, only Wargny et al.'s study had adjusted glucose tolerance levels [13].

Recent reports have shown that NAFLD can occur in individuals who are not obese and have a normal BMI. These individuals are labeled as "lean NAFLD" or "nonobese NAFLD" [14]. Ye et al. reported that in the general population (comprising individuals with and without NAFLD), 12.1% of people had non-obese NAFLD and 5.1% had lean NAFLD [15]. Young et al. reported that the prevalence of lean NAFLD in the general population was 11.2% worldwide and 12% in Asia [16]. Metabolic risk factors associated with insulin resistance are relevant for non-obese and obese NAFLD [17]. In addition, lean or non-obese NAFLD is a risk factor for the development of diabetes [18–22]. Approximately 25% and 40% of all NAFLD cases are lean and non-obese, respectively [15,16]. Therefore, focusing only on obesity may miss patients with NAFLD and those at a high risk of developing diabetes.

Obesity, as defined by BMI, is a common risk factor for both NAFLD and diabetes. Since BMI is a component of FLI, the confounding effect of obesity must be considered when assessing the incidence of diabetes using FLI. Many previous studies have argued that FLI is a predictor of the development of new diabetes mellitus [7,10–13,23–29]. However, there have been no studies on FLI and the development of diabetes in non-obese individuals.

The aim of this study was to assess the risk of developing diabetes in FLI-NAFLD after strict classification of blood glucose status among the Japanese population. Therefore, we limited the subjects in this study to those with prediabetes, as assessed by both FPG and HbA1c criteria. Additionally, we aimed to evaluate whether or not FLI-NAFLD is associated with the risk of developing diabetes considering obesity and prediabetes levels.

## Methods

### Study design and data source

The present study was a retrospective study performed using the Japanese health check-up and administrative claims databases from FY2015 to FY2018. Data were obtained from one health insurance association, comprising annual health check-up and claims data collected from all prefectures in Japan other than Tokyo. The database comprises information on the age, sex, diagnosis, prescriptions, medical procedures, and regions.

### Study subjects

Subjects were diagnosed with prediabetes using both the HbA1c criterion and FPG. Eligible subjects for this study were those who (1) underwent the annual health check-up at FY2015 and had data available; (2) had no missing data for weight, height, WC, HbA1c, FPG, TG, or GGT according to the questionnaire of the use of antidiabetics in the health check-up at FY2015 and FY2018; (3) had no cardiovascular disease, chronic kidney disease, or stroke according to the questionnaire of the health check-up at FY2015; (4) did not drink alcohol every day, with daily alcohol consumption not exceeding 20 g of ethanol according to the questionnaire of the health check-up at FY2015; (5) had no claims of ICD-10 codes for B18 (chronic viral hepatitis), C22 (malignant neoplasm of the liver and intrahepatic bile ducts), K743 (primary biliary cirrhosis), or K754 (autoimmune hepatitis) at FY2015; (6) had no outlier data for HbA1c or WC at FY2015 and 2018; (7) did not have diabetes (HbA1c ≥6.5% or FPG ≥126 mg/dL or use of antidiabetics) at FY2015; and (8) did not have normoglycemia (HbA1c <5.7% and FPG <100 mg/dL) at FY2015. Subjects who met all eligibility criteria are shown in Fig 1.

The definition of "prediabetes" is shown in Fig 2. According to ADA criteria, prediabetics with HbA1c >6.0% are considered to be at a high risk and require aggressive intervention and vigilant follow-up [8]. Therefore, in this study, HbA1c >6.0% was defined as "high-risk prediabetes" and HbA1c ≤6.0% as "standard prediabetes." In the original Bedogni et al. study, FLI ≥60 was suggested to rule in FLD, but in Asians, the cutoff value for FLI in NAFLD diagnosed by ultrasonography is often approximately 30 [30,31]. Therefore, FLI ≥30 was used to define FLI-NAFLD in this study. The International Obesity Task Force recommended the lower cut-offs of BMI ≥23kg/m$^2$ for overweight, and ≥25.0kg/m$^2$ for obese for Asian people, according to the risk for type 2 diabetes and hypertension [32]. According to Japanese guidelines, obesity is defined as a BMI of 25 kg/m$^2$ or higher [33]. Therefore, BMI≥25 kg/m$^2$ was used to define obesity in this study. The patterns were grouped into four groups according to the following criteria: BMI <25 kg/m$^2$ with FLI <30 as "non-obese without FLI-NAFLD"; BMI ≥25 kg/m$^2$ with FLI <30 as "obese without FLI-NAFLD"; BMI <25 kg/m$^2$ with FLI ≥30 as "non-obese with FLI-NAFLD"; and BMI ≥25 kg/m$^2$ with FLI ≥30 as "obese with FLI-NAFLD."

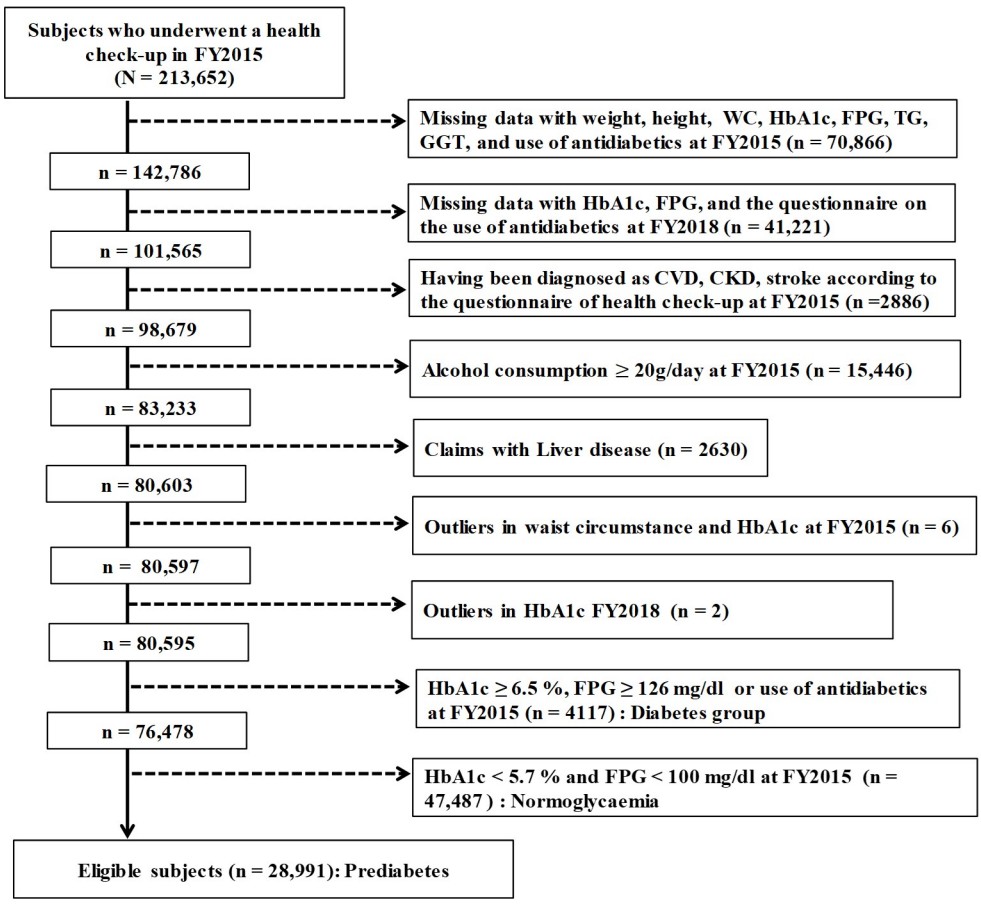

**Fig 1. Flow of eligible subjects.**

## Outcomes and variables

The primary outcome was the development of diabetes mellitus at the time of the 2018 health check-up. Diabetes mellitus was defined as HbA1c $\geq$6.5%, FPG $\geq$126 mg/dl, or use of antidiabetic medication in the questionnaire.

For background variables, the age, sex, FLI, high-density lipoprotein cholesterol (HDL-c), low-density lipoprotein cholesterol (LDL-c), systolic blood pressure (SBP), HbA1c (based on National Glycohemoglobin Standardization Program units), comorbidities (hypertension, hyperlipidemia, diabetes mellitus based on self-administered questionnaire), eating habits, smoking habits, and physical activities at FY2015 were extracted from the database. The FLI

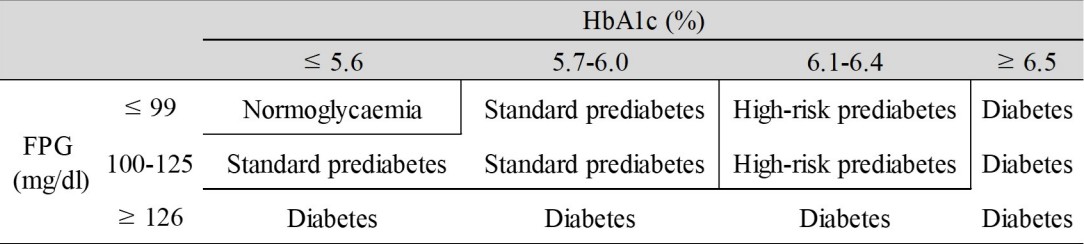

| | | HbA1c (%) | | | |
|---|---|---|---|---|---|
| | | ≤ 5.6 | 5.7-6.0 | 6.1-6.4 | ≥ 6.5 |
| FPG (mg/dl) | ≤ 99 | Normoglycaemia | Standard prediabetes | High-risk prediabetes | Diabetes |
| | 100-125 | Standard prediabetes | Standard prediabetes | High-risk prediabetes | Diabetes |
| | ≥ 126 | Diabetes | Diabetes | Diabetes | Diabetes |

**Fig 2. Definition of prediabetes.**

score was calculated as follows:

$$FLI = \left\{ \frac{e^{0.953 \times \log(TG) + 0.139 \times BMI + 0.718 \times \log(\gamma - GTP) + 0.053 \times WC - 15.745}}{1 + \left(e^{0.953 \times \log(TG) + 0.139 \times BMI + 0.718 \times \log(\gamma - GTP) + 0.053 \times WC - 15.745}\right)} \right\} \times 100$$

## Statistical analysis

For descriptive statistics of baseline characteristics, mean (SD) or median (IQR) were calculated for continuous variables, and frequency and percentage were calculated for categorical variables for each group. The chi-squared test or analysis of variance was performed to compare groups for each aggregated background.

Univariate and multivariate analyses were performed to evaluate the background and risk of developing diabetes. Multivariate adjusted logistic regression models were then applied to evaluate the association between the development of diabetes and the combination of BMI and FLI. Multivariate analyses with three models were performed to calculate odds ratios (ORs) and their 95% confidence intervals (CIs). Models were adjusted for age (categorized into three age groups: 39–49, 50–59, and 60–71 years) and sex for Model-1; Model-1 plus FPG, SBP, HDL, and LDL for Model-2; Model-2 plus smoking, eating habit, physical activities, adequate sleeping, weight change within 1 year, and age at 20 for Model-3.

Yang et al. reported cutoff value for FLI that differed by sex (35 for males and 20 for females) [31], so we also performed a sensitivity analysis using the Yang et al. cutoff values.

The dose-dependent analysis between FLI and incident diabetes was performed. The FLI was categorized into seven FLI groups (0– <10, 10– <20, 20– <30, 30– <40, 40– <50, 50– <60, and ≥60) and multivariate adjusted logistic regression models were then applied. The model was adjusted for age, sex, BMI, FPG, SBP, HDL, LDL, smoking, eating habit, physical activities, adequate sleeping, weight change within 1 year, and age at 20.

To evaluate the fitness of the model, we performed the lack-of-fit test [34].

All tests were two-tailed, and the significance level was set to 0.05. For the statistical analysis, R version 3.63 (R Core [Team 2020] R: A language and environment for statistical computing. R Foundation for Statistical Computing, Vienna, Austria. URL https://www.R-project.org/) and JMP® version 15.0 (SAS Institute Inc., Cary, NC, USA) was used.

Results are reported in accordance with the recommendations of the Strengthening the Reporting of Observational Studies in Epidemiology (STROBE) [35].

## Ethics

In this study, only anonymized data were used and we had no access to personal information. The Teikyo University Ethical Review Board for Medical and Health Research Involving Human Subjects approved this study after due ethical consideration (approval No.:18-200-3).

## Results

### Study population

Of the 213,652 beneficiaries, 28,991 eligible subjects were extracted from the database. For patients with normoglycemia, standard prediabetes, and high-risk prediabetes, the cumulative incidences of new-onset diabetes were 0.19% (92/47,487 cases), 2.96% (770/26,014 cases), and 26.1% (777/2977 cases), respectively.

### Baseline characteristics

Descriptive analyses for baseline characteristics of the eligible subjects are shown in Table 1. Analyses of lifestyle factors, such as eating habits, physical activities, and daily sleeping, are provided in Table in S1 Table. Every variable significantly differed among the four groups.

### Risk of new-onset diabetes in patients with standard prediabetes and high-risk prediabetes

In the multivariate analysis, significant factors associated with an increased risk of developing diabetes in patients with standard prediabetes were FLI $\geq$30, BMI $\geq$25 kg/m$^2$, female sex, current smoking, weight increase >3 kg within 1 year, and no walking or exercise for at least 1 h/day (Table 2). In patients with high-risk prediabetes, FLI $\geq$30, current smoking, and weight gain of 10 kg or more since age 20 were significant factors. In addition, there was no interaction between obesity and FLI-NAFLD (*P*-value for interaction: 0.69).

The results of the multivariate adjusted logistic regression analysis in the three models are shown in Table 3 and Fig 3. Patients with obesity with FLI-NAFLD, whether standard or high-risk prediabetes had a significantly higher risk of developing diabetes than non-obese subjects without FLI-NAFLD in all three models. In Model-3, non-obese patients with FLI-NAFLD and patients with obesity without FLI-NAFLD were at a higher risk in patients with standard prediabetes, with ORs of 1.44 (95% CI: 1.09–1.92; *P* = 0.011) and 1.79 (95% CI: 1.21–2.65; *P* = 0.003), respectively, but were not found in patients with high-risk prediabetes.

The results of sensitivity analysis with a cutoff value of FLI of 35 for males and 20 for females are shown in Table 4, and descriptive analyses for baseline characteristics of the eligible subjects are shown in Table in S2 Table. The results were not remarkable changed, but one difference was that in high-risk prediabetes, the non-obese with FLI-NAFLD group had a significant OR from 1.42 (0.99–2.03) to 1.68 (1.18–2.39).

The results of the dose-dependent analysis between FLI and incident diabetes is shown in Fig 4. There was a dose-dependent relationship between FLI and the development of diabetes in both standard and high-risk prediabetes, which was more marked in standard prediabetes.

## Discussion

The present study showed that FLI-NAFLD was an independent risk factor for the development of diabetes in the middle-aged Japanese population. The association between FLI-NAFLD and diabetes incidence differs with obesity and prediabetes levels. Patients with obesity with FLI-NAFLD was a higher risk factor for the development of diabetes than non-obese patients without FLI-NAFLD.

In the non-obese population, FLI-NAFLD was an independent risk factor for incident diabetes in patients with standard prediabetes (OR, 1.44; *P* = 0.011). Patients with high-risk prediabetes had a moderately increased risk of developing diabetes (OR: 1.42; *P* = 0.054). These results suggest that FLI is an effective tool for screening high-risk prediabetic individuals for the development of diabetes, even in non-obese individuals, particularly those with HbA1c $\leq$6.0%.

Concerning prediabetic levels, the population of "high-risk prediabetes" was only approximately one-ninth of that of "standard prediabetes." However, the number of new cases of diabetes was almost the same. HbA1c levels in 90% of patients with prediabetes were below 6.0% (standard prediabetes). In this majority population, 70% of new cases of diabetes were complicated with FLI-NAFLD, which accounted for only 38.6% of this population. The diabetes incidence was 3.0% in the standard prediabetes group and 26.1% in the high-risk prediabetes

**Table 1. Baseline characteristics of eligible subjects.**

| | Non-obese without FLI-NAFLD | | Non-obese with FLI-NAFLD | | Obese without FLI-NAFLD | | Obese with FLI-NAFLD | |
|---|---|---|---|---|---|---|---|---|
| | n = 14,676 | | n = 3230 | | n = 1291 | | n = 6817 | |
| **Standard prediabetes (HbA1c; 5.7–6.0%)** | | | | | | | | |
| Sex (Male), n (%) | 8808 | 60.0% | 2919 | 90.4% | 721 | 55.8% | 5441 | 79.8% |
| Age (year), n (%) | | | | | | | | |
| 39–49 | 7227 | 49.2% | 1410 | 43.7% | 683 | 52.9% | 3521 | 51.7% |
| 50–59 | 7181 | 48.9% | 1754 | 54.3% | 587 | 45.5% | 3188 | 46.8% |
| 60–71 | 268 | 1.8% | 66 | 2.0% | 21 | 1.6% | 108 | 1.6% |
| Current smoker, n (%) | 1504 | 10.3% | 648 | 20.1% | 115 | 8.9% | 1246 | 18.3% |
| Metabolic syndrome* | 107 | 0.7% | 464 | 14.4% | 62 | 4.8% | 2329 | 34.2% |
| FLI | 9.9 | (5.1–17.2) | 41.3 | (34.8–51.6) | 22.8 | (18.1–26.2) | 59.5 | (44.9–75.7) |
| BMI (kg/m$^2$) | 21.6 | (20.0–22.9) | 23.8 | (22.24.4) | 25.7 | (25.3–26.4) | 27.4 | (26.29.4) |
| WC (cm) | 77.6 | ±6.2 | 85.1 | ±4.2 | 86.7 | ±4.1 | 94.4 | ±7.4 |
| SBP (mmHg) | 115.7 | ±15.7 | 122.5 | ±15.5 | 121 | ±15.3 | 126.8 | ±15.6 |
| DBP (mmHg) | 72.5 | ±11.4 | 78.6 | ±11.3 | 75.9 | ±11.4 | 81 | ±11.6 |
| Laboratory tests | | | | | | | | |
| HbA1c (%) | 5.7 | ±0.2 | 5.6 | ±0.2 | 5.7 | ±0.2 | 5.7 | ±0.2 |
| FPG (mg/dl) | 97.9 | ±8.4 | 101.5 | ±7.7 | 99.3 | ±7.9 | 101.3 | ±8.1 |
| AST (U/L) | 19 | (17–23) | 23 | (20–28) | 19 | (17–23) | 24 | (20–30) |
| ALT (U/L) | 17 | (13–22) | 28 | (21–39) | 18 | (14–24) | 31 | (22–41) |
| GGT (U/L) | 21 | (16–30) | 54 | (36–87) | 19 | (15–25) | 40 | (28–61) |
| LDL-C (mg/dl) | 123.6 | ±28.6 | 135.8 | ±31.5 | 124.8 | ±27.9 | 136.4 | ±30.5 |
| HDL-C (mg/dl) | 64 | (55–75) | 52 | (44–61) | 58 | (51–68) | 50 | (44–58) |
| TG (mg/dl) | 76 | (57–100) | 154 | (118–206) | 73 | (59–89) | 132 | (100–181) |
| Hemoglobin (mg/dl) | 14.0 | ±1.6 | 15.1 | ±1.1 | 14.0 | ±1.6 | 15.0 | ±1.3 |
| | **Non-obese without FLI-NAFLD** | | **Non-obese with FLI-NAFLD** | | **Obese without FLI-NAFLD** | | **Obese with FLI-NAFLD** | |
| | n = 1080 | | n = 377 | | n = 133 | | n = 1387 | |
| **High-risk prediabetes (HbA1c; 6.1–6.5%)** | | | | | | | | |
| Sex (Male), n (%) | 599 | 55.5% | 327 | 86.7% | 61 | 45.9% | 1061 | 76.5% |
| Age (year), n (%) | | | | | | | | |
| 39–49 | 379 | 35.1% | 121 | 32.1% | 57 | 42.9% | 611 | 44.1% |
| 50–59 | 664 | 61.5% | 239 | 63.4% | 71 | 53.4% | 751 | 54.1% |
| 60–71 | 37 | 3.4% | 17 | 4.5% | 5 | 3.8% | 25 | 1.8% |
| Current smoker, n (%) | 159 | 14.7% | 92 | 24.4% | 10 | 7.5% | 280 | 20.2% |
| Metabolic syndrome* | 27 | 2.5% | 85 | 22.6% | 8 | 6.0% | 694 | 50.0% |
| FLI | 11.8 | (6.1–19.0) | 41.7 | (35.5–52.0) | 22.9 | (18.7–26.8) | 66.8 | (51.0–81.9) |
| BMI (kg/m$^2$) | 21.9 | (20.4–23.1) | 23.8 | (23.1–24.5) | 25.6 | (25.3–26.4) | 28.2 | (26.5–30.5) |
| WC (cm) | 78.3 | ±6.0 | 85.3 | ±4.1 | 86.2 | ±4.9 | 96.3 | ±8.5 |
| SBP (mmHg) | 116.1 | ±15.3 | 121.9 | ±13.9 | 123.2 | ±14.0 | 127.5 | ±15.0 |
| DBP (mmHg) | 72.7 | ±11.0 | 77.8 | ±10.7 | 76.1 | ±10.1 | 81.4 | ±11.4 |
| Laboratory tests | | | | | | | | |
| HbA1c (%) | 6.2 | ±0.1 | 6.2 | ±0.1 | 6.2 | ±0.1 | 6.2 | ±0.1 |
| FPG (mg/dl) | 101.2 | ±10.4 | 106.7 | ±9.5 | 103.3 | ±9.7 | 106.5 | ±9.5 |
| AST (U/L) | 20 | (17–23) | 24 | (19–29) | 21 | (17–25) | 26 | (21–34) |
| ALT (U/L) | 18 | (14–23) | 29 | (22–42) | 20 | (16–28) | 36 | (25–55) |
| GGT (U/L) | 22 | (16–30) | 53 | (36–81) | 20 | (15–26) | 44 | (31–66) |
| LDL-C (mg/dl) | 128.9 | ±30.2 | 136.8 | ±32.5 | 134 | ±34.2 | 137.9 | ±30.9 |

(*Continued*)

**Table 1.** (Continued)

| | | | | | | | | |
|---|---|---|---|---|---|---|---|---|
| HDL-C (mg/dl) | 62 | (53–73) | 50 | (44–57) | 59 | (51–66) | 49 | (44–57) |
| TG (mg/dl) | 80 | (59–107) | 157 | (124–210) | 75 | (62–98) | 138 | (103–188) |
| Hemoglobin (mg/dl) | 13.7 | ±1.7 | 14.9 | ±1.2 | 14 | ±1.4 | 15 | ±1.4 |

AST, aspartate aminotransferase; ALT, alanine aminotransferase.

* According to Japanese diagnostic criteria [36].

group, showing a nine-fold difference between the two groups. The prevalence of obesity significantly differed between standard prediabetes (31%) and high-risk prediabetes (51%) groups. In patients with high-risk prediabetes, neither obesity (BMI $\geq$25 kg/m$^2$) nor FLI-NAFLD alone was a risk factor, but both were strong risk factors (OR = 1.73). In particular, obesity alone was not a risk factor for the development of diabetes (OR = 0.89). A quarter of this population developed diabetes whereas half was obese. HbA1c >6.0% itself is a strong risk factor, and since obesity accounts for half of the population, BMI $\geq$25 kg/m$^2$ is probably not a risk factor.

In the present study, 1,547 new patients with diabetes were identified from 28,911 patients with prediabetes. Of the patients with newly diagnosed diabetic, 37.8% had a BMI <25 kg/m$^2$. Furthermore, the incidence of diabetes was higher in non-obese patients with FLI-NAFLD than in those without FLI-NAFLD (6.0% vs. 4.5%).

The difference in the risk of diabetes in patients with obesity and FLI-NAFLD can be explained by "lipid spillover." Asians, particularly East Asians, have a lower capacity for fat storage in subcutaneous adipose tissue than in other ethnic groups [37]. Therefore, lipid spillover, in which free fatty acid (FFA) overflows from adipose tissue, is thought to be more likely. Lipid spillover may result in the accumulation of ectopic fat, such as fatty liver, which may lead to insulin resistance. Kadowaki et al. evaluated fat distribution, adipose tissue insulin resistance, and skeletal muscle insulin resistance in non-obese Japanese men [38]. Even among non-obese individuals, visceral and hepatic fat accumulations were observed in some individuals, with various accumulation patterns. Even in the absence of visceral fat accumulation, muscle insulin resistance (metabolic disturbance) was observed in the presence of fatty liver, whereas no insulin resistance was observed in the absence of fatty liver, even in the presence of visceral fat accumulation. Non-obese prediabetes should not be neglected. In such cases, FLI is the best screening method because it is non-invasive, inexpensive, and can be calculated using health check-up data.

In this study, the NAFLD population was dominantly male. The first reason for this was that the subjects were 19,937 men and 9054 women, with men accounting for 69% of the total. Second, the prevalence of NAFLD, defined as FLI $\geq$ 30, was 49% (9748) in men and 23% (2063) in women. The FLI values were significantly higher in males than in females, which seemed to cause a gender difference in the prevalence of FLI-NAFLD. Though there was concern that the gender difference in the prevalence of FLI-NAFLD might be a selection bias, the results of sensitivity analysis using a cutoff value of FLI of 35 for males and 20 for females also showed no significant difference. Rather, the odds ratio of non-obese FLI-NAFLD in high-risk prediabetes changed from 1.42 (0.99–2.03) to 1.68 (1.18–2.39), which was significant. This result further supported the finding that a high FLI is a risk factor for developing diabetes, even in non-obese individuals. In addition, for gender risk, there seemed to be a strong confounding of FLI, laboratory values (especially SBP, FPG, HDL), and lifestyle (especially current smoking). After adjusting for these confounding factors, biologically speaking, women were at higher risk of developing diabetes than men in the present study.

**Table 2. Univariate and multivariate adjusted logistic regression models for the incidence of new-onset diabetes.**

| | Univariate model | | | Multivariate model* | | |
|---|---|---|---|---|---|---|
| | OR | 95% CI | *P*-value | OR | 95% CI | *P*-value |
| **Standard prediabetes (HbA1c; 5.7–6.0%)** | | | | | | |
| FLI ≥ 30 | 3.94 | (3.37–4.60) | < .0001 | 1.40 | (1.10–1.77) | 0.006 |
| BMI ≥ 25 | 3.57 | (3.08–4.14) | < .0001 | 1.68 | (1.35–2.09) | < .0001 |
| Sex (male) | 1.99 | (1.66–2.39) | < .0001 | 0.76 | (0.61–0.94) | 0.013 |
| Age | | | | | | |
| 39–49 | reference | | | | | |
| 50–59 | 1.09 | (0.95–1.27) | 0.224 | 1.03 | (0.87–1.21) | 0.762 |
| 60–71 | 1.31 | (0.80–2.16) | 0.280 | 1.25 | (0.71–2.19) | 0.443 |
| Life stile | | | | | | |
| Current smoking | 1.99 | (1.68–2.36) | < .0001 | 1.81 | (1.48–2.21) | < .0001 |
| Weight change of more than ±3 kg within 1 year | 1.96 | (1.67–2.29) | < .0001 | 1.37 | (1.15–1.64) | < .001 |
| Weight gain of 10 kg or more since aged at 20 | 2.71 | (2.31–3.18) | < .0001 | 1.10 | (0.91–1.35) | 0.327 |
| Light exercise of at least 30 minutes per session | 0.93 | (0.77–1.12) | 0.445 | 0.99 | (0.80–1.22) | 0.900 |
| 1 hour per day of walking or exercise in daily life | 1.07 | (0.91–1.30) | 0.406 | 1.22 | (1.01–1.48) | 0.038 |
| Walking speed is faster | 0.88 | (0.75–1.03) | 0.100 | 0.86 | (0.72–1.02) | 0.079 |
| Eating dinner within 2 hours before bedtime | 1.26 | (1.07–1.48) | 0.005 | 1.04 | (0.87–1.25) | 0.632 |
| Eating midnight snack | 1.01 | (0.83–1.23) | 0.896 | 0.94 | (0.76–1.16) | 0.537 |
| Skipping breakfast | 1.54 | (1.25–1.91) | < .0001 | 1.00 | (0.79–1.26) | 0.993 |
| Eating speed | | | | | | |
| Slow | reference | | | | | |
| Normal | 1.43 | (0.96–2.14) | 0.082 | 1.07 | (0.70–1.65) | 0.741 |
| Fast | 2.29 | (1.53–3.43) | < .0001 | 1.34 | (0.87–2.06) | 0.185 |
| Adequate sleeping | 1.08 | (0.92–1.26) | 0.350 | 1.02 | (0.86–1.20) | 0.863 |
| **High-risk prediabetes (HbA1c; 6.1–6.5%)** | | | | | | |
| FLI ≥ 30 | 2.40 | (2.00–2.88) | < .0001 | 1.56 | (1.15–2.11) | 0.004 |
| BMI ≥ 25 | 2.00 | (1.69–2.37) | < .0001 | 1.13 | (0.86–1.49) | 0.371 |
| Sex (male) | 1.78 | (1.47–2.15) | < .0001 | 0.89 | (0.69–1.15) | 0.382 |
| Age | | | | | | |
| 39–49 | reference | | | | | |
| 50–59 | 0.93 | (0.79–1.10) | 0.421 | 0.87 | (0.71–1.07) | 0.189 |
| 60–71 | 0.58 | (0.33–1.04) | 0.066 | 0.47 | (0.22–1.00) | 0.051 |
| Life stile | | | | | | |
| Current smoking | 1.62 | (1.32–1.98) | < .0001 | 1.41 | (1.10–1.82) | 0.008 |
| Weight change of more than ±3 kg within 1 year | 1.76 | (1.45–2.12) | < .0001 | 1.57 | (1.26–1.95) | < .0001 |
| Weight gain of 10 kg or more since aged at 20 | 1.64 | (1.36–1.97) | < .0001 | 0.89 | (0.70–1.13) | 0.337 |
| Light exercise of at least 30 minutes per session | 0.99 | (0.80–1.23) | 0.919 | 1.02 | (0.70–1.33) | 0.865 |
| 1 hour per day of walking or exercise in daily life | 0.92 | (0.75–1.10) | 0.420 | 0.99 | (0.78–1.26) | 0.955 |
| Walking speed is faster | 0.96 | (0.80–1.14) | 0.623 | 1.00 | (0.81–1.23) | 0.978 |
| Eating dinner within 2 hours before bedtime | 1.20 | (0.99–1.46) | 0.007 | 1.05 | (0.84–1.32) | 0.644 |
| Eating midnight snack | 0.94 | (0.75–1.18) | 0.621 | 0.95 | (0.74–1.23) | 0.692 |
| Skipping breakfast | 1.47 | (1.12–1.93) | 0.006 | 1.22 | (0.89–1.67) | 0.206 |
| Eating speed | | | | | | |
| Slow | reference | | | | | |
| Normal | 1.26 | (0.28–0.83) | 0.279 | 1.17 | (0.73–1.87) | 0.508 |
| Fast | 1.38 | (0.90–2.11) | 0.142 | 1.04 | (0.65–1.68) | 0.859 |
| Adequate sleeping | 0.91 | (0.76–1.08) | 0.281 | 0.86 | (0.70–1.06) | 0.150 |

OR, odds ratio; CI, 95% confidence interval.

* Adjusted for age, sex, FPG, SBP, HDL, LDL, smoking, eating habit, physical activities, adequate sleeping, weight change within year and since aged at 20.

**Table 3. Multivariate adjusted logistic regression model for the incidence of new-onset diabetes by the combination pattern of obesity and FLI-NAFLD.**

| | New-onset diabetes FY2018 | | Model 1† | | | Model 2‡ | | | Model 3§ | | |
|---|---|---|---|---|---|---|---|---|---|---|---|
| | n | % | OR | 95% CI | P-value* | OR | 95% CI | P-value* | OR | 95% CI | P-value* |
| Standard prediabetes (n = 26,014) | | | | | | | | | | | |
| Non-obese without FLI-NAFLD (n = 14,676) | 190 | 1.29% | reference | | | reference | | | reference | | |
| Non-obese with FLI-NAFLD (n = 3230) | 112 | 3.47% | 2.45 | (1.92–3.12) | < .0001 | 1.60 | (1.24–2.06) | < .001 | 1.44 | (1.09–1.92) | 0.011 |
| Obese without FLI-NAFLD (n = 1291) | 38 | 2.94% | 2.36 | (1.66–3.36) | < .0001 | 1.85 | (1.29–2.65) | < .001 | 1.79 | (1.21–2.65) | 0.003 |
| Obese with FLI-NAFLD (n = 6817) | 430 | 6.31% | 4.81 | (4.03–5.73) | < .0001 | 2.80 | (2.30–3.41) | < .0001 | 2.36 | (1.85–3.01) | < .0001 |
| High-risk prediabetes (n = 2977) | | | | | | | | | | | |
| Non-obese without FLI-NAFLD (n = 1080) | 117 | 16.39% | reference | | | reference | | | reference | | |
| Non-obese with FLI-NAFLD (n = 377) | 106 | 28.12% | 1.79 | (1.35–2.37) | < .0001 | 1.25 | (0.92–1.71) | 0.155 | 1.42 | (0.99–2.03) | 0.054 |
| Obese without FLI-NAFLD (n = 133) | 26 | 19.55% | 1.29 | (0.81–2.04) | 0.278 | 0.99 | (0.61–1.61) | 0.973 | 0.89 | (0.51–1.57) | 0.693 |
| Obese with FLI-NAFLD (n = 1387) | 468 | 33.74% | 2.40 | (1.96–2.93) | < .0001 | 1.60 | (1.27–2.03) | < .0001 | 1.73 | (1.29–2.32) | < .001 |

OR, odds ratio; CI, 95% confidence interval.

* P-values are derived from multivariate logistic regression model.

† Adjusted for age and sex.

‡ Adjusted for age, sex, FPG, SBP, HDL and LDL.

§ Adjusted for age, sex, FPG, SBP, HDL, LDL, smoking, eating habit, physical activities, adequate sleeping, weight change within year and since aged at 20.

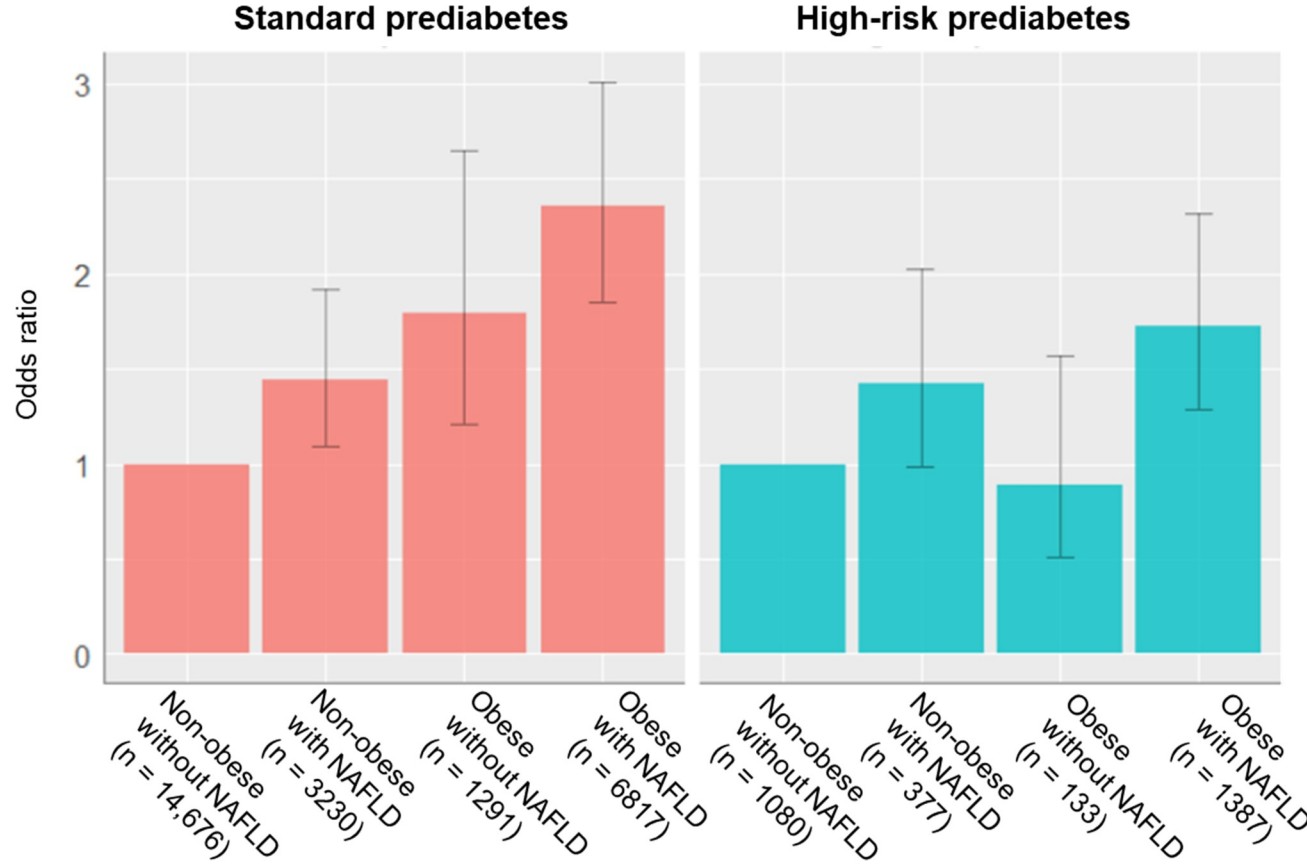

**Fig 3. Risk of developing new diabetes for each combination pattern of FLI-NAFLD and obesity.** The bars represent each odds ratio for non-obese patients without FLI-NAFLD, and error bars represent 95% CI of the odds ratio.

**Table 4. Sensitivity analysis of the multivariate adjusted logistic regression model for the incidence of new-onset diabetes by the combination pattern of obesity and FLI-NAFLD, with a cutoff value of FLI of 35 for men and 20 for women.**

|  | New-onset diabetes FY2018 | | Model 1[†] | | | Model 2[‡] | | | Model 3[§] | | |
|---|---|---|---|---|---|---|---|---|---|---|---|
|  | n | % | OR | 95% CI | *P*-value* | OR | 95% CI | *P*-value* | OR | 95% CI | *P*-value* |
| Standard prediabetes (n = 26,014) | | | | | | | | | | | |
| Non-obese without FLI-NAFLD (n = 14,961) | 209 | 1.40% | reference | | | reference | | | reference | | |
| Non-obese with FLI-NAFLD (n = 2945) | 93 | 3.16% | 2.17 | (1.70–2.79) | < .0001 | 1.42 | (1.09–1.84) | 0.009 | 1.35 | (1.01–1.81) | 0.042 |
| Obese without FLI-NAFLD (n = 1336) | 45 | 3.37% | 2.26 | (1.63–3.14) | < .0001 | 1.72 | (1.23–2.41) | 0.002 | 1.58 | (1.09–2.30) | 0.015 |
| Obese with FLI-NAFLD (n = 6772) | 423 | 6.25% | 4.49 | (3.79–5.32) | < .0001 | 2.65 | (2.19–3.20) | < .0001 | 2.31 | (1.83–2.92) | < .0001 |
| High-risk prediabetes (n = 2977) | | | | | | | | | | | |
| Non-obese without FLI-NAFLD (n = 1104) | 180 | 16.30% | reference | | | reference | | | reference | | |
| Non-obese with FLI-NAFLD (n = 353) | 103 | 29.18% | 2.03 | (1.53–2.69) | < .0001 | 1.57 | (1.15–2.14) | 0.004 | 1.68 | (1.18–2.39) | 0.004 |
| Obese without FLI-NAFLD (n = 132) | 27 | 20.45% | 1.22 | (0.77–1.92) | 0.392 | 1.09 | (0.67–1.76) | 0.735 | 1.06 | (0.62–1.83) | 0.826 |
| Obese with FLI-NAFLD (n = 1388) | 467 | 33.65% | 2.46 | (2.02–3.00) | < .0001 | 1.73 | (1.37–2.18) | < .0001 | 1.81 | (1.35–2.41) | < .0001 |

OR, odds ratio; CI, 95% confidence interval.

* P-values are derived from multivariate logistic regression model.

† Adjusted for age and sex.

‡ Adjusted for age, sex, FPG, SBP, HDL and LDL.

§ Adjusted for age, sex, FPG, SBP, HDL, LDL, smoking, eating habit, physical activities, adequate sleeping, weight change within year and since aged at 20.

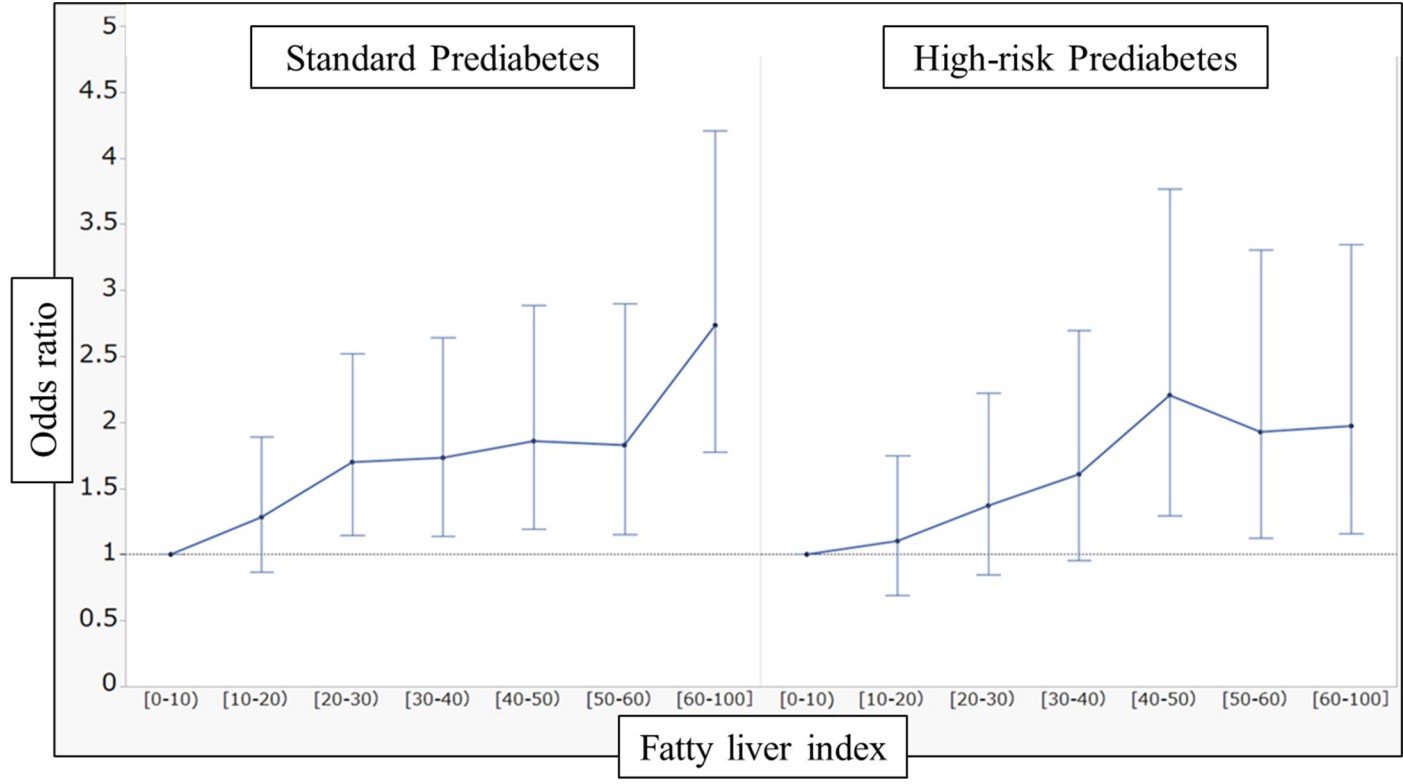

**Fig 4. Graph of FLI and risk of incident diabetes.** The dots represent each odds ratio for the categories classified by FLI values, and error bars represent 95% CI of the odds ratio.

The strength of this study was that it was the largest cohort study (n = 28,911) on the relationship between FLI and the risk of diabetes incidence among patients with prediabetes. Therefore, we could stratify and limit our analysis by glucose tolerance levels and obesity, which allowed us to prevent and address confounding factors. In addition, study subjects were civil servants from Japan, and more than 90% of insurance beneficiaries in this study had undergone a specific health checkup every year, leading to a small sampling bias.

This study had several limitations. First, we did not perform the 2-h oral glucose tolerance test, so the prevalence of diabetes might be underestimated. Second, the optimal cutoff FLI score for predicting NAFLD has been controversial in Asia; however, most cutoff scores reported by validation studies on FLI using ultrasonography in Asia are approximately 30. Third, the study population was limited to a single insurance member. Although it is a sampling bias, it is the same occupation and is less affected by bias due to the socioeconomic status. Finally, unmeasured confounders, such as a family history of diabetes, were not included in this study.

In Japan, to prevent MetS, all public health insurers are obliged to provide specific health checkups and health guidance. Obesity based on WC and BMI is a mandatory criterion to select subjects for health guidance. Therefore, there is a lack of evaluation of lifestyle-related diseases and health guidance for non-obese people. Even in non-obese individuals, the risks of developing insulin resistance and diabetes increase if they have fatty liver, so it is necessary to improve their lifestyle. FLI can be calculated using only health checkup test items and may be effective in identifying individuals at a high risk for lifestyle-related diseases, particularly diabetes, among non-obese individuals.

In conclusion, by assessing FLI in combination with obesity, an association between FLI and those at high risk of developing diabetes in a middle-aged Japanese population was observed. Validating these results, it is desirable to develop a screening tool to effectively identify people at high risk of developing diabetes from the population (especially non-obese prediabetes) who are apparently at low health risk and are unlikely to be targeted for health guidance.

## Supporting information

**S1 Table. Analysis of lifestyle factors.**
(TIF)

**S2 Table. Baseline characteristics of the subjects in a sensitivity analysis with the cutoff value of FLI as 35 for males and 20 for females.**
(TIF)

## Acknowledgments

We thank Honyaku Center Inc. (https://www.honyakucenter.jp/) for editorial support.

## Author Contributions

**Conceptualization:** Atsushi Kitazawa.

**Data curation:** Shotaro Maeda.

**Formal analysis:** Atsushi Kitazawa, Shotaro Maeda.

**Investigation:** Atsushi Kitazawa, Shotaro Maeda.

**Methodology:** Atsushi Kitazawa.

**Project administration:** Atsushi Kitazawa.

**Software:** Shotaro Maeda.

**Supervision:** Yoshiharu Fukuda.

**Visualization:** Shotaro Maeda.

**Writing – original draft:** Atsushi Kitazawa.

**Writing – review & editing:** Yoshiharu Fukuda.

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
