## [Decision Letter · Decision Letter 0]

24 May 2021

PONE-D-21-12023

Fatty liver index as a predictive marker for the development of diabetes: a retrospective cohort study using Japanese health check-up data

PLOS ONE

Dear Dr. Kitazawa,

Thank you for submitting your manuscript to PLOS ONE. After careful consideration, we feel that it has merit but does not fully meet PLOS ONE’s publication criteria as it currently stands. Therefore, we invite you to submit a revised version of the manuscript that addresses the points raised during the review process.

We look forward to receiving your revised manuscript.

Kind regards,

Ming-Lung Yu, MD, PhD

Academic Editor

PLOS ONE

Journal Requirements:

3. This is a retrospective study observational study. As such, we do not feel that any conclusions on the effectiveness of FLI as a screening tool may be supported; thus, we ask that you revise the text (especially Conclusions) to avoid unsupported statements.

4. Thank you for stating the following in the Financial Disclosure section:

[The authors received no specific funding for this work.].   

We note that one or more of the authors are employed by a commercial company: Kyorin Pharmaceutical

Reviewers' comments:

Reviewer's Responses to Questions

**Comments to the Author**

1. Is the manuscript technically sound, and do the data support the conclusions?

Reviewer #1: Yes

Reviewer #2: Partly

2. Has the statistical analysis been performed appropriately and rigorously? 

Reviewer #1: Yes

Reviewer #2: Yes

3. Have the authors made all data underlying the findings in their manuscript fully available?

Reviewer #1: Yes

Reviewer #2: Yes

4. Is the manuscript presented in an intelligible fashion and written in standard English?

Reviewer #1: Yes

Reviewer #2: Yes

5. Review Comments to the Author

Reviewer #1: Non-alcoholic fatty liver disease (NAFLD) is an increasingly cause of chronic liver disease. Current studies have been shown that NAFLD contributed substantially to the development of insulin resistance and type 2 diabetes mellitus (DM). In this study, Kitazawa et al. investigated the predictive value of fatty liver index for the development of DM in patients with prediabetes. They found that high FLI is a risk factor for the development of diabetes. Although the results were clinically interesting, several points need be critically addressed.

1. In this study, obesity was defined as body mass index (BMI) ≥25 kg/m2. However, the cutoff for obesity is 23.0 kg/m2 in Asia-Pacific countries, including Japan. The authors may perform subgroups analysis by BMI <23 versus ≥23 kg/m2.

2. The authors should provide the data that how many patients have metabolic syndrome. The correlation between and the development of DM should be taken into consideration.

3. In this study, the cutoff value for FLI in NAFLD was set at 30. The authors should analyze the dose-dependent relationship between FLI and incidence of DM.

4. The cutoff value for FLI in NAFLD were different between male and female gender. Usually, the cut-off value was lower in females than in males. Regardless of gender, the cutoff value for FLI in NAFLD was set at 30 in this study. The authors should explain this important issue.

5. The inclusion criteria (8) did not have normoglycemia (HbA1c �6.5% or FPG �126 mg/dL or use of antidiabetics) at FY2015. Patients with normoglycemia should have HbA1c < 5.7% or FPG <100 mg/dL. Please correct.

Reviewer #2: Authors addressed to find a tool to screen and identify subjects who are at high risk of progressing to diabetes mellitus and FLI maybe the choice of the tool. There are many tools and scores to identify subjects who are progressing to diabetes mellitus, however, maybe FLI is more suitable for Asian population.

Here are some concerns, the most important one is the selection bias. We noted among the population of non-obese with FLI-NAFLD and obese with FLI-NAFLD, both in HbA1c 5.7-6.0% and HbA1c 61.-6.4%, the majority of gender is male.

Major consideration:

Table 1: the gender ratio of non-obese with FLI-NAFLD and obese with FLI-NAFLD were almost male (90.4% and 79.8%), why?

Minor consideration:

1. Table 1, current smoker: is the smokers almost male? What is the proportion?

2. Table 2: the male risk is 1.99 in univariate model, and 0.76 in multivariate model, can you explain it?

3. How to define walking and eating speed is faster?

4. Page 7, line 116: (3) had no outlier data: how to define “outlier” data?

6. PLOS authors have the option to publish the peer review history of their article (what does this mean?). If published, this will include your full peer review and any attached files.

Reviewer #1: No

Reviewer #2: No

---

## [Author Response · Author response to Decision Letter 0]

30 Jun 2021

We appreciate your comments on our manuscript. The comments have helped us significantly improve the paper.

---

## [Decision Letter · Decision Letter 1]

20 Jul 2021

PONE-D-21-12023R1

Fatty liver index as a predictive marker for the development of diabetes: a retrospective cohort study using Japanese health check-up data

PLOS ONE

Dear Dr. Kitazawa,

Thank you for submitting your manuscript to PLOS ONE. After careful consideration, we feel that it has merit but does not fully meet PLOS ONE’s publication criteria as it currently stands. Therefore, we invite you to submit a revised version of the manuscript that addresses the points raised during the review process.

We look forward to receiving your revised manuscript.

Kind regards,

Ming-Lung Yu, MD, PhD

Academic Editor

PLOS ONE

Journal Requirements:

Reviewers' comments:

Reviewer's Responses to Questions

**Comments to the Author**

1. If the authors have adequately addressed your comments raised in a previous round of review and you feel that this manuscript is now acceptable for publication, you may indicate that here to bypass the “Comments to the Author” section, enter your conflict of interest statement in the “Confidential to Editor” section, and submit your "Accept" recommendation.

Reviewer #1: All comments have been addressed

Reviewer #2: All comments have been addressed

2. Is the manuscript technically sound, and do the data support the conclusions?

Reviewer #1: Yes

Reviewer #2: Yes

3. Has the statistical analysis been performed appropriately and rigorously? 

Reviewer #1: Yes

Reviewer #2: Yes

4. Have the authors made all data underlying the findings in their manuscript fully available?

Reviewer #1: Yes

Reviewer #2: No

5. Is the manuscript presented in an intelligible fashion and written in standard English?

Reviewer #1: Yes

Reviewer #2: Yes

6. Review Comments to the Author

Reviewer #1: This revised manuscript is much improved and all previous comments were responded on point-to-point basis. I have no additional comments.

Reviewer #2: Dear authors, you had replied my concerns and did a good work about this manuscript.

According to author's reply, you use a new cutoff value of FLI of 35 for males and 20 for females and it seems have better results in this study. "the odds ratio of non-obese FLI-NAFLD in high-risk prediabetes changed from 1.42 (0.99-2.03) to 1.68 (1.18-2.39), which was significant."

Can you upload these differences in fully supplement tables and why don't you use the new cutoff value instead of the original one?

Why the original enrolled subjects are male dominant? 19,937 in male v.s. 9054 in female?

7. PLOS authors have the option to publish the peer review history of their article (what does this mean?). If published, this will include your full peer review and any attached files.

Reviewer #1: No

Reviewer #2: No

---

## [Author Response · Author response to Decision Letter 1]

6 Aug 2021

Reply to Reviewer #2

1.According to author's reply, you use a new cutoff value of FLI of 35 for males and 20 for females and it seems have better results in this study. "the odds ratio of non-obese FLI-NAFLD in high-risk prediabetes changed from 1.42 (0.99-2.03) to 1.68 (1.18-2.39), which was significant. Can you upload these differences in fully supplement tables and why don't you use the new cutoff value instead of the original one?

Reply.

First, because the protocol was initially determined with FLI 30 as the cutoff value. Secondly, there is no common consensus on the cutoff value of FLI in Asia. Therefore, we thought it should be easier to use the FLI as a screening tool in health examination results. Therefore, we used a common cutoff value for men and women.

The results of sensitivity analysis with a cutoff value of FLI of 35 for males and 20 for females are shown in Table 4, and descriptive analyses for baseline characteristics of the eligible subjects are shown in Table in S2 Table.

2.Why the original enrolled subjects are male dominant? 19,937 in male v.s. 9054 in female?

Reply.

This is because the gender distribution of the employers in this insurance association is significantly more male.

---

## [Decision Letter · Decision Letter 2]

31 Aug 2021

Fatty liver index as a predictive marker for the development of diabetes: a retrospective cohort study using Japanese health check-up data

PONE-D-21-12023R2

Dear Dr. Kitazawa,

We’re pleased to inform you that your manuscript has been judged scientifically suitable for publication and will be formally accepted for publication once it meets all outstanding technical requirements.

Kind regards,

Ming-Lung Yu, MD, PhD

Academic Editor

PLOS ONE

Additional Editor Comments (optional):

Reviewers' comments:

Reviewer's Responses to Questions

**Comments to the Author**

1. If the authors have adequately addressed your comments raised in a previous round of review and you feel that this manuscript is now acceptable for publication, you may indicate that here to bypass the “Comments to the Author” section, enter your conflict of interest statement in the “Confidential to Editor” section, and submit your "Accept" recommendation.

Reviewer #1: All comments have been addressed

Reviewer #2: All comments have been addressed

2. Is the manuscript technically sound, and do the data support the conclusions?

Reviewer #1: Yes

Reviewer #2: Yes

3. Has the statistical analysis been performed appropriately and rigorously? 

Reviewer #1: Yes

Reviewer #2: Yes

4. Have the authors made all data underlying the findings in their manuscript fully available?

Reviewer #1: Yes

Reviewer #2: Yes

5. Is the manuscript presented in an intelligible fashion and written in standard English?

Reviewer #1: Yes

Reviewer #2: Yes

6. Review Comments to the Author

Reviewer #1: This revised manuscript is much improved and all previous comments were responded on point-to-point basis. I have no additional comments.

Reviewer #2: They showed cutoff value of FLI 35 for male and 20 for female in table 4 and detailed characteristics in S2 table, and they had fully responded my opinions. Good work!

7. PLOS authors have the option to publish the peer review history of their article (what does this mean?). If published, this will include your full peer review and any attached files.

Reviewer #1: No

Reviewer #2: No

---

## [Editor Report · Acceptance letter]

6 Sep 2021

PONE-D-21-12023R2 

Fatty liver index as a predictive marker for the development of diabetes: a retrospective cohort study using Japanese health check-up data 

Dear Dr. Kitazawa:

I'm pleased to inform you that your manuscript has been deemed suitable for publication in PLOS ONE. Congratulations! Your manuscript is now with our production department. 

Kind regards, 

on behalf of

Dr. Ming-Lung Yu 

Academic Editor

PLOS ONE